# Package Design Thermal Optimization for Metal-Oxide Gas Sensors by Finite Element Modeling and Infra-Red Imaging Characterization

**DOI:** 10.3390/ma16186202

**Published:** 2023-09-14

**Authors:** Serguei Stoukatch, Francois Dupont, Philippe Laurent, Jean-Michel Redouté

**Affiliations:** Microsys Laboratory, Department of Electrical Engineering and Computer Science (Institut Montefiore), University of Liège, 4000 Liège, Belgium; fff.dupont@uliege.be (F.D.); p.laurent@uliege.be (P.L.); jean-michel.redoute@uliege.be (J.-M.R.)

**Keywords:** thermal modeling, thermal management of electronic packages, finite element modeling (FEM), MOX sensor packaging, microassembly

## Abstract

We designed a 3D geometrical model of a metal-oxide gas sensor and its custom packaging and used it in finite element modeling (FEM) analysis for obtaining temperature and heat flux distribution. The 3D computer simulation, performed with GetDP software (version 3.5.0, 13 May 2022), accurately predicted the temperature distribution variation across the entire assembly. Knowing the temperature variation and the location of the hot spots allowed us to select the best electrical interconnect method and to choose the optimal materials combination and optimal geometry. The thermal modeling also confirmed the need to use a low thermal conductivity material to insulate the MOX sensor since the latter is heated to its operational temperature of 250 °C. For that purpose, we used the in-house formulated xerogel–epoxy composite of thermal conductivity of 0.108 W m^−1^ K^−1^, which is at least 30% less compared to the best-in-class among commercially available materials. Based on the 3D FEM outputs, we designed, assembled, and characterized a fully functional packaged MOX gas sensor in several configurations. We measured the temperature distribution on all parts of the MOX gas sensor assembly using a thermal imaging infrared (IR) microscope. The results of 3D FEM are in good agreement with the temperature distribution obtained by the non-contact IR thermal characterization.

## 1. Introduction

Thermal management concerns are present in many types of electronic packages [1]. Excessive heat can not only affect the system performance but also cause reliability concerns [2] and, in the worst-case scenario, can damage thermal sensitive parts of the electronic assembly. Miniaturization, as well as the requirement for cost reduction, implies more compact and better-optimized packages [3] in terms of geometry and material used. Thermal modeling can answer such questions.

There are numerous efforts on thermal management [4] in different fields of electronic packages. For example, for high-performance computing systems [5], it is important to maintain performance while components are affected by thermally related issues caused by excessive heat generation. Continuous efforts for miniaturization resulted in the vertical integration of 2D integrated circuits (ICs) in a 3D-stacked assembly, which led to a further increase in energy density, difficulties with heat dissipation, and causes additional thermal management challenges [6]. Among the mentioned challenges, it is important to identify the hot spots [7] locations to prevent heat concentration in localized areas and to evacuate excessive heat from the system. Thermal management problems are also actual for traditional packaging of multi-chip modules [8]. Such issues are also widely reported in system-level thermal management for power electronics [9], where the use of passive (such as heat spreaders and heat sinks) and/or active (such as fan or liquid-based) cooling systems is paramount. Automotive electronics [10], including underhood applications [11], are also known for harsh environments including excessive heat generation, temperature variation, and vibrations.

In many cases, commercially available materials are not able to cope with the increasing demand dictated by thermal management requirements. In response to that, researchers have developed new materials or modified existing ones. For example, in [12], boron nitride nanosheets (BNNs) were described to enhance the thermal conductivity of polymer-based packaging materials. In [13], the required low thermal conductivity was achieved by formulating an epoxy–xerogel composite. Reference [14] reported modified properties of graphene using the so-called laser-modified graphene process, which can be used in various applications.

Nowadays, there are different commercially available multiphysics software to model and simulate varieties of physical phenomena. Most of them are also suitable for thermal-related issues.

We review below the most known commercially available software. It is important to note that the order of mentioning does not imply any ranking between them nor specific preference among users. Furthermore, this list does not pretend to be exhaustive.

We start the review with Cadence [15], which provides tools for thermal analysis, including the modeling of steady-state and transient temperature profiles. It offers adaptable analytical heat transfer models with a combination of a numerical computation approach that can be used for different thermal problems including 2D- and 3D-stacked ICs.

Comsol [16] successfully commercialized a multiphysics simulation platform that analyzes thermal aspects in varieties of micro- and macro- objects and investigates thermal designs and the impact of heat loads. The multiphysics platform allows 3D finite element modeling of temperature fields and heat fluxes, considering all three modes of heat transfer: conduction, convection, and radiation. Ansys [17] offers thermal management and fluid flow solutions for many types of electronic design applications, including high-power electronics, enabling comprehensive electronics simulation and design capabilities. Infineon [18] uses SPICE and SABER simulation models for their power–electronic systems that contain a dynamic link between electrical and thermal component descriptions. MathWorks [19] offers basic thermal blocks and modeling techniques that are suitable for modeling multi-domain physical systems including varieties of thermal sensors, thermal elements, and heat sources. M3d [20] is a fully integrated finite element modeler, mesh, solver, and post-processor. QuickField 6.6 (6 December 2021) [21] simulation software offers a finite element analysis package and coupled multi-field analysis for multiple physical problems including heat transfer and stress design simulation. ROHM’s [22] home page provides thermal models of different electronic architectures.

An alternative for the multiphysics software to address the thermal problem is to replace the thermal models with their electrical equivalents in the form of RC network models [23], such as the Cauer and Foster networks [24]. Such thermal networks are typically used to build a compact model for quick calculations, and this approach has been used in solutions that are provided commercially by several companies. For example, Mathworks [25] supports Cauer thermal modeling for studying heat transfer via multiple layers of a semiconductor module. Siemens [26] developed corresponding software to simulate electronic cooling and to perform thermal analysis for electronic circuits. This software comprises extensive libraries, automated data handling, a tailored and stable solver technology, automatic model calibration and parametric analyses, and other features that typically are not available in open-source software.

The mentioned commercial tools are not free, and depending on the user agreement, are usually fully or partially charged. The objective of this short review is to summarize the most established commercial modeling options to the reader. However, while much less numerous, there are also open-access software alternatives that allow us to achieve similar results. Within the scope of this work, we used GetDP [27], a free finite element solver developed at the University of Liege.

Traditionally [28], MOX gas sensors are packaged in metal–ceramic discreet packages that are capable of withstanding high operational temperatures. High operating temperatures of 250 °C and above are necessary to activate the sensitive chemo-resistive layer of the MOX sensor. For example, according to [29], the maximum sensitivity for carbon dioxide (CO_2_) detection of zinc oxide (ZnO) thick film-based gas sensor is achieved at 300 °C. The heating is performed by a micro-heater integrated into the sensor. Typically [30], the packaged MOX sensor is mounted on the printed circuit board (PCB) as a through-hole component where the relatively long leads are inserted through holes on the PCB and soldered. Because of weak thermal coupling between the package and the PCB, the PCB remains at a relatively low temperature, and the effect of heated MOX on the PCB is typically neglected. In the case of directly mounting the MOX sensor of the PCB [31], the situation is drastically different, and thermal analyses must be performed in order to ensure that the PCB temperature remains in the acceptable range. The results of thermal analysis are the distribution of temperature over the parts of the assembly, as well as the identification of hot spots. This latest configuration, which allows for reducing the cost and the volume of the assembly, is the one studied in this paper.

This paper is structured as follows: The MOX sensor is briefly described in Section 2, including its structure and other important features. In Section 3, we present the 3D model of the packaged MOX sensor. In Section 4, we present the results of 3D finite element modeling. In Section 5, we validate the modeling results by thermography using an IR camera. Finally, we draw conclusions in Section 6.

## 2. MOX Gas Sensor: Package Description and Construction

In this work, we studied two packaging alternatives for MOX gas sensors, using two different interconnection schemes. In the conductive adhesive (CA) case, the electrical interconnection between the MOX sensor bond pads and the PCB is performed using CA tracks, while in the wire bonding (WB) case, the interconnection is achieved by conventional gold wire bonds. The CA tracks have a cross-sectional area of about 0.4 × 0.8 mm^2^ and are applied by dispensing silver-filled conductive adhesive epoxy ABLESTIK 84-1LMIT1, from Henkel. The wire bonds are made by using a 25 µm diameter gold bonding wire. The PCB is a standard FR4 type PCB, with a glass transition temperature (*T_g_*) of 150 °C, a thermal conductivity of 0.3 W m^−1^ K^−1^, and is 1 mm thick.

The MOX sensor is made of ceramic alumina. It is 0.5 mm thick and has lateral dimensions of 2.9 × 2.9 mm^2^. For its permanent fixation to the PCB, we used an in-house [13] specially formulated thermal interface material (TIM) with a thermal conductivity of 0.108 W m^−1^ K^−1^ [32], which comprises 22.7% xerogel by weight. This thermally insulating adhesive aims to protect the PCB from the heat generated by the sensor, its operational temperature being as high as 250 °C. More details on the sensor and PCB geometry, as well as package construction, can be found in [33]. The photographs of the CA and WB MOX sensor from frontside, backside, and front-angle view are presented in Figure 1.

## 3. 3D Modeling

Among the available multiphysics software packages that are potentially capable of performing the required thermo-analyses, we selected the GetDP open-source software [27]. GetDP is a software that was developed as an open-source solution to address discrete physical problems. The environment is open to a couple of numerous physical problems and numerical methods including finite element and boundary methods. GetDP, unlike commercial software, is fully free of charge.

In our work using GetDP software, we developed a full 3D model of the packaged MOX gas sensor. The 3D model geometry comprises three main functional parts: the bare MOX sensor die, the PCB, and the electrical and thermomechanical interconnections between them, shortly named DIE, PCB, TRACKS, and TIM. The 3D geometrical model of the packaged MOX gas sensor was meshed and solved using the GetDP software. The mesh was automatically performed based on a characteristic length. This length equals 125 µm on all the packaged sensors except at the corners of the PCB where it equals 250 µm, so the size of the element is slightly larger when approaching the external part of the PCB, but it is 125 µm on all the sensitive parts of the sensor (DIE, TRACKS, TIM, and PCB directly in contact with the TIM). In the air volume, starting from the external border of the packaged sensor, the characteristic length is 500 µm.

The triangles in most volumes are equilateral, hence the tetrahedrons based on these triangles are regular. However, in gold wires, the triangles are not equilateral and have a sharper shape. This is because one side of the triangles is constrained by the width of the wires while along the length of the wire, the characteristic length of 125 µm is retained.

Each functional part comprises several materials that have different thermal conductivities that are listed and summarized in Table 1.

We studied the packaged sensor thermal behavior by applying a steady-state thermal model with conductive heat transfer taken into account in all volumes, including the air (assuming constant thermal properties), and convective and radiative heat transfer taken into account on all external surfaces of the assembly. For $h_{conv}$ and $h_{rad}$ (convective and radiative transfer coefficients, respectively), we used $h_{conv}$ = 20 W K^−1^ m^−2^ and $h_{rad}=\varepsilon \sigma$ where ε=0.75 and σ is the Stefan–Boltzmann constant. This is an approximation, as these coefficients are neither constant nor uniform over the different parts of the assembly. Convective heat transfer depends on the surfaces’ orientation, while radiative heat transfer depends on the emissivity coefficient of the different materials (DIE, TRACKS, TIM, and PCB). This approximation is, however, acceptable as we verified that it does not impact a significant extent on the comparative results between wire bonding and CA, where the main contributing factor is the conductive heat transfer.

## 4. Results of the 3D Simulation

Figure 2 presents 3D temperature distribution in steady-state on the top and bottom of the gas sensor with CA (a, b) and WB (c, d) interconnections. In each case, the power supplied to the sensor heater is adjusted to reach a sensor operating temperature of 250 °C at the bottom of the surface die (where the resistive heater is located). Modeling results are obtained for the pure epoxy (a,c,e,g) and the xerogel–epoxy composite (b,d,f,h).

Using the same 3D model, we also obtained the 3D heat flux distribution. In Figure 3, we illustrate the heat flux distribution across the gas sensor assembly with xerogel–epoxy composite for CA case (a) and WB case (b) interconnection.

The heat flux is indicated by small arrows giving its direction and by the color corresponding to its amplitude.

The heat flux is two orders of magnitude larger in the WB case (1 × 10^7^–3 × 10^7^ W/m²) compared to the CA case (1 × 10^5^–3 × 10^5^ W/m²), but as the cross-sectional area for the WB is much smaller than for CA, it results that the average power extracted from the die is approximate of the same order of magnitude. Indeed, the injected power required to keep the bottom surface of the die at 250 °C is 850 mW in the CA case and 570 mW in the WB case. From the corresponding 3D plots (Figure 3a,b), it can be seen that in the WB case, this heat is diverted from the die/TIM/PCB top surface to go directly on the copper pins while, in the CA case, the drained heat causes the elevation of the temperature of the TIM and the PCB in that area because the CA tracks are in direct contact with the top surface. This explains the larger average temperature of the TIM and the PCB in the CA case.

Figure 4 presents temperature distribution across the top and the bottom of the gas sensor with CA and WB interconnection for a temperature at the bottom surface of the die of 250 °C. Modeling results are obtained for the standard epoxy and for the xerogel–epoxy composite.

On the bottom side of the TIM below the center of the die, the temperature in the CA tracks case configuration is 228 °C for a target temperature of 250 °C while, in the WB case configuration, it decreases to 210 °C. The temperature profiles are the same on the top and the bottom side outside the copper pins that act as a thermal channel across the thickness of the PCB.

The use of WB decreases strongly (several tens of °C) the average temperature of the PCB (red and blue curves in comparison to black and green curves). It can be seen in both figures corresponding to the two faces (top of the die and backside of the PCB). In addition, the use of the xerogel–epoxy composite instead of the pure epoxy further decreases the temperature of the PCB because of its lower thermal conductivity; a larger thermal gradient takes place in the thickness of the TIM layer. The target temperature of 250 °C at the backside of the die is reached for different values of injected power: 907 mW for the CA case with pure epoxy versus 570 mW for the best case (WB case with the xerogel–epoxy composite).

As a result of modeling, we obtained a temperature distribution across all parts of the MOX gas sensor assembly, including the gas sensor, PCB, TIM (pure epoxy and xerogel–epoxy composite), etc. We identified the temperature range and hotspots correspondingly. The temperature ranges as a function of the package configuration (CA and WB case, pure epoxy and xerogel–epoxy composite) are presented in Table 2.

## 5. Modeling Validation

The most straightforward way to validate the results obtained via 3D modeling is to directly measure the temperature distribution across the assembly and compare it with the modeling results. There are different techniques [35] to measure temperature distribution across the microassemblies, the most common among them [36] is either to use contact probes, such as thermocouples and thermistors, or non-contact methods using, for instance, IR thermography. For microassemblies, such as the studied miniaturized MOX sensors, thermal characterization using an IR camera is the most suitable method, taking into account the small features of the components. In [37], the authors demonstrated the use of the IR camera to measure temperature distribution accurately and in a repeatable way on miniaturized MOX sensors.

In the present work, we performed temperature distribution measurements across the assembly by using a commercially available high-resolution thermal imaging system Sentris IS640 manufactured by OPTOTHERM, USA [38]. The system comprises a long wave infrared (IR) (7–14 µm range) detector. The IR system ensures high-temperature measurement accuracy up to 1 °C and a spatial lateral resolution of 5 µm. The system is equipped with a heating work holder and is fully controlled by an external PC with proprietary software. The experimental setup for non-contact temperature measurements across the MOX gas sensor assembly using an IR camera is presented in Figure 5.

The thermal imaging microscope Sentris IS640 (Optotherm, Sewickley, PA, USA) was configured with a 20 µm objective that provides a sufficient field of view of 12.8 × 9.6 mm^2^ to observe the entire assembly. The heating element on the backside of the sensor was electrically connected to the circuit to reach the sensor operating temperature of 250 °C.

Using the above-mentioned setup, the observed temperature distribution in WB and CA packaged sensors is reported in Figure 6. The sensors operate in steady-state at a temperature of 250 °C.

As was demonstrated and discussed in detail in [33], because of the different materials (PCB, TIM, ceramic, CA tracks, and WB) of which the packaged sensor is made and their different corresponding emissivities, the temperature readings obtained directly by IR camera must be corrected in order to obtain the actual temperature. In this experiment, we applied the earlier developed procedure for temperature correction to obtain the actual temperature of the specific assembly parts. The actual temperature range, minimum, and maximum on TIM and PCB correspondingly obtained by the IR thermography together with the related results obtained by 3D FEM modeling are listed in Table 3.

The temperature distribution across the package obtained by 3D FEM modeling is in good agreement with the results obtained by non-contact IR thermography.

The injected power (Q) from an external power source to the sensor heater required to reach a temperature of 250 °C on the bottom surface of the die as measured by the IR microscope was compared with the corresponding power obtained by 3D modeling. The comparison is presented in Table 4.

The experimental data for the injected power is almost the same for the WB case (580 mW versus 570 mW obtained by modeling), while for the CA case, the difference between the experimental and modeled value is about 28%. This discrepancy can be explained as follows. The boundary conditions applied for the interface used the averaged and constant values. Our model uses approximated data for the physical and geometrical properties of CA and WB interconnections. These physical properties can be experimentally obtained to improve the accuracy of the simulation. The more substantial underestimation of the heat losses for the CA interconnection can be explained by significantly larger contact surfaces for the CA tracks compared to the gold wires that cause additional heat dissipation because of the direct contact with the surrounding air. The corresponding thermal properties used in the model were averaged, and in the case of the CA interconnection, the discrepancy between the averaged and the actual values due to its larger contact surface is more significant.

Another possible explanation is that the real electrical conductivity of the obtained CA tracks is lower than the expected value deduced from the data sheet (generally based on fully optimized curing conditions).

## 6. Conclusions

We developed a full 3D model with the open-source GetDP software and utilized it for the finite element modeling of temperature distribution across the gas sensor. The 3D computer simulation accurately predicted the temperature variation. This approach allowed us to select the best electrical interconnect method and choose an optimal materials combination and assembly geometry. Based on the modeling, we assembled and characterized MOX gas sensors. We measured the temperature distribution across the assembly with the thermal imaging IR microscope Sentris IS640 to validate the results of the 3D modeling. The results of the modeling are in good agreement with the temperature distribution obtained by the thermal imaging microscopy. The thermally optimized electronic package can be suitable for assembly varieties of MOX gas sensors [39] that require high operating temperatures. The in-house formulated xerogel–epoxy composite with very low thermal conductivity that played a key role in the presented novel packaging approach can be useful for packaging carbon dot (CD)-based sensors. The fluorescence response of CDs decreases with increasing temperature from 20 °C to 80 °C [40], and the temperature of CDs should be kept below 80 °C. The CD sensors show a fast response time and high selectivity and sensitivity in several practical applications such as heavy-metal sensing [40], varieties of biological fluids [41], acetone in human fluids [42], and antibacterial applications [40].

## Figures and Tables

**Figure 1 materials-16-06202-f001:**
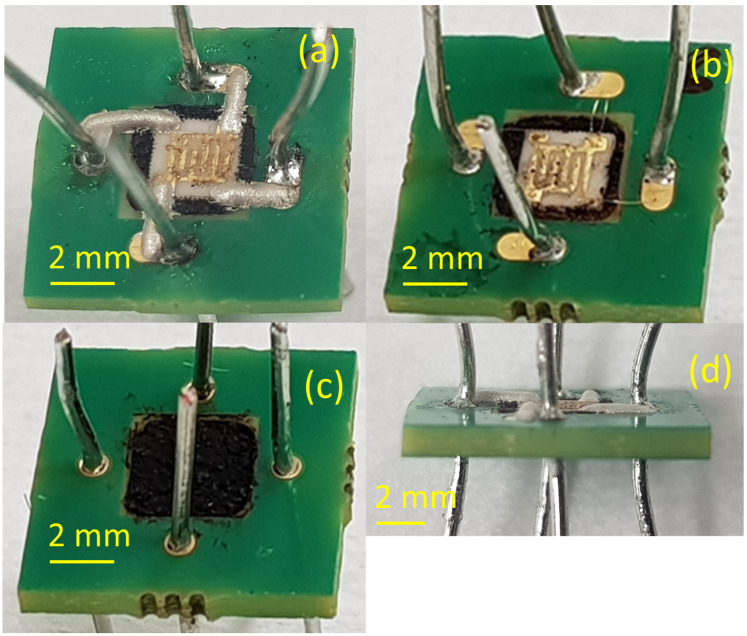
Photographs of the frontside of the CA (**a**) and WB (**b**) sensor, backside (**c**) of the sensor, and front-angle view (**d**) on the CA sensor.

**Figure 2 materials-16-06202-f002:**
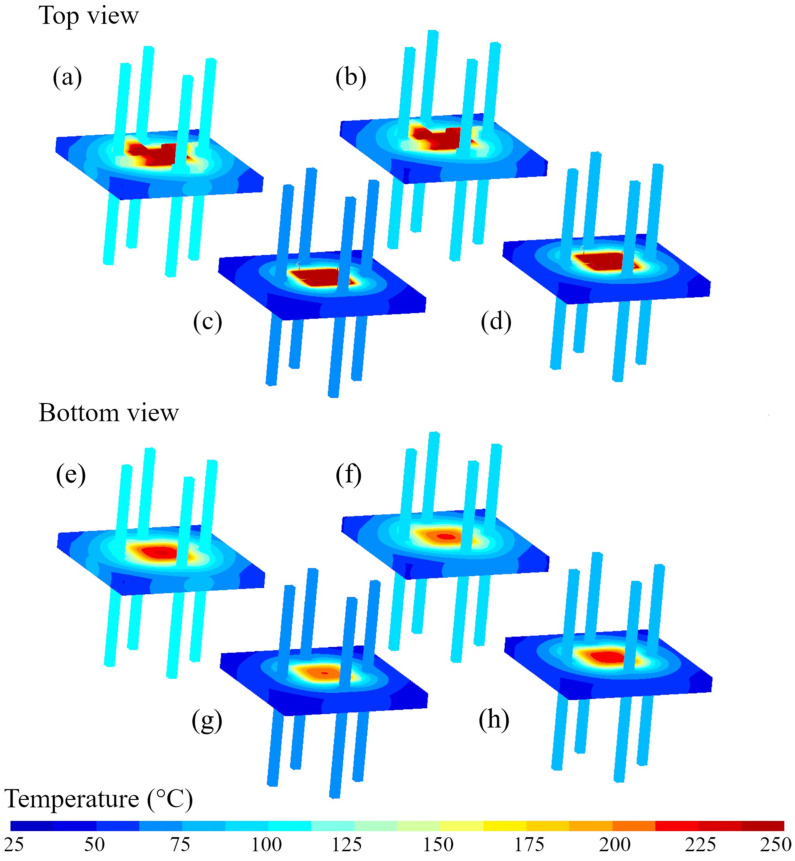
The 3D temperature distribution on the top (**a**,**b**) and the bottom (**e**,**f**) of the gas sensor with CA (**a**,**b**) and on the top (**c**,**d**) and the bottom (**g**,**h**) of the gas sensor with WB interconnection for a temperature at the bottom surface of the die of 250 °C. Modeling results are obtained for the standard epoxy (**a**,**c,e,g**) and for the epoxy–xerogel composite (**b**,**d**,**f**,**h**).

**Figure 3 materials-16-06202-f003:**
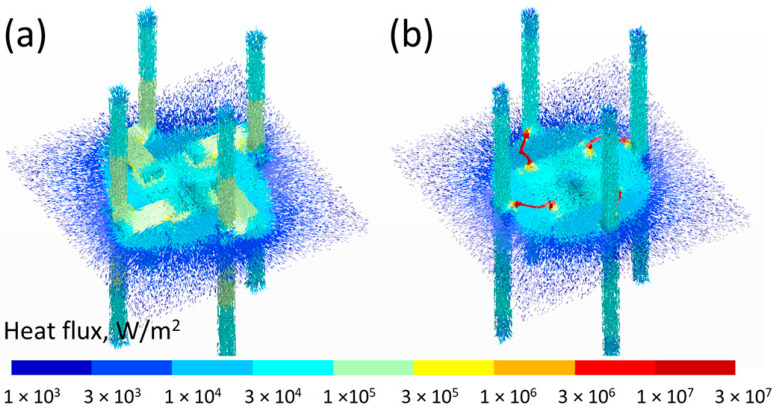
The 3D heat flux distribution across the gas sensor assembly with xerogel–epoxy composite for CA case (**a**) and WB case (**b**) interconnection.

**Figure 4 materials-16-06202-f004:**
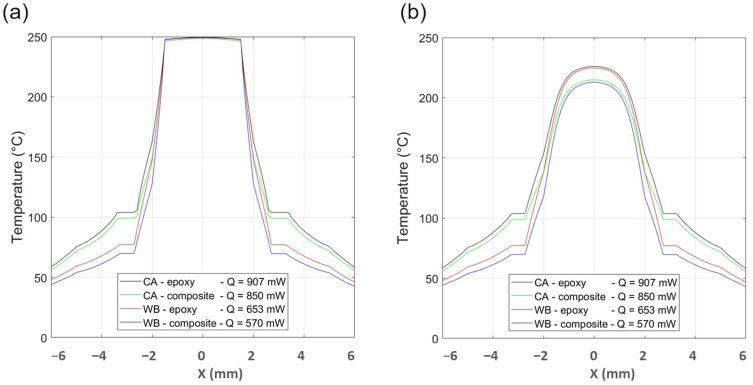
Temperature distribution across the top (**a**) and the bottom (**b**) of the gas sensor assembly with CA and WB interconnection with epoxy and xerogel–epoxy composite. Q is the injected power required to reach a temperature of 250 °C on the bottom surface of the die.

**Figure 5 materials-16-06202-f005:**
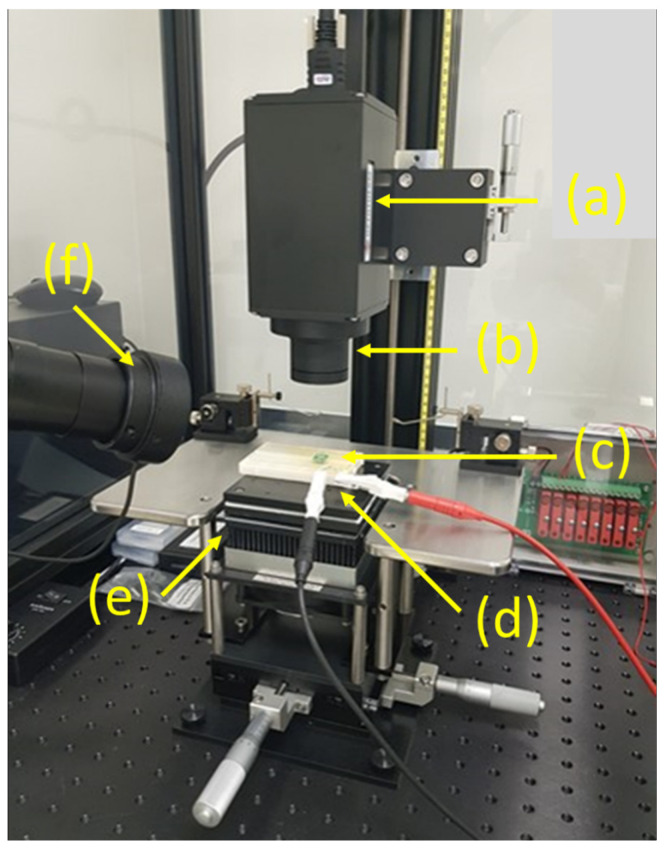
Experimental setup for non-contact temperature measurements across the assembly MOX gas sensor using IR camera: (a) the IR camera, (b) the objective, (c) the sample, (d) the power supply, (e) the worktable with 3D micro-positioner, and (f) the visible light camera.

**Figure 6 materials-16-06202-f006:**
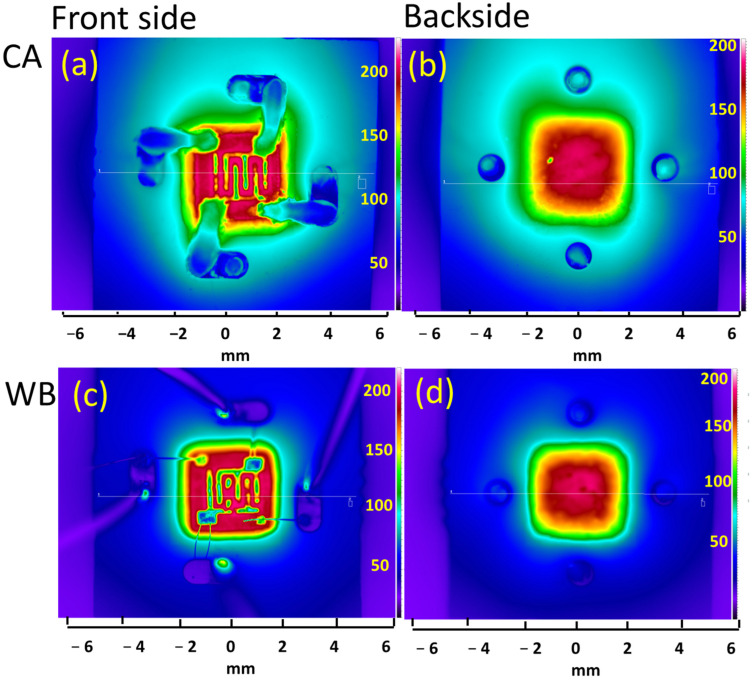
Temperature distribution of the CA (**a**,**b**) and WB (**c**,**d**) sensor at the front (**a**,**c**) and backside (**b**,**d**) of assembly for a setup temperature of 250 °C. © [2021] IEEE. Reprinted (partially) with permission from [37].

**Table 1 materials-16-06202-t001:** Thermal conductivities of all materials used in the 3D model.

Material	k (W K^−1^ m^−1^)
Sensor die (alumina ceramic)	25 [33]
PCB	0.30 [33]
Thermal insulating material (TIM)	
xerogel–epoxy composite	0.108 [32]
pure epoxy	0.168 [32]
Interconnection method:	
CA tracks: (CA case)	3.6 [34]
Gold wires (WB case)	317 [34]
Copper leads	380 [34]
Air	
at 20 °C	0.0259 [34]
at 250 °C	0.0414 [34]

**Table 2 materials-16-06202-t002:** Temperature ranges across the assembly parts TIM and PCB as a function of the package configuration.

Case	TIM	TIM *T*_min_/*T*_max_ (°C)	PCB *T*_min_/*T*_max_ (°C)
CA	pure epoxy	125/250	49/187
CA	xerogel–epoxy	114/250	47/184
WB	pure epoxy	108/250	41/150
WB	xerogel–epoxy	92/250	38/131

**Table 3 materials-16-06202-t003:** Temperature ranges were obtained by modeling and experimentally by IR thermography at an operating temperature of 250 °C across the assembly parts TIM and PCB as a function of the package configuration.

Case	TIM *T*_min_/*T*_max_ (°C)		PCB *T*_min_/*T*_max_ (°C)	
	Modeling	Experimental	Modeling	Experimental
CA	112/250	104/231	54/180	68/151
WB	87/250	92/222	42/117	49/115

**Table 4 materials-16-06202-t004:** Injected power (Q) required to reach a temperature of 250 °C on the bottom surface of the die obtained by modeling and experimentally by IR thermography.

Case	*Q* (mW)	
	Modeling	Experimental
CA	850	1180
WB	570	580

## Data Availability

The data presented in this study are available in the document (Figure 1, Figure 2, Figure 3, Figure 4, Figure 5 and Figure 6, Table 1, Table 2, Table 3 and Table 4).

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
