# Peer review of "Package Design Thermal Optimization for Metal-Oxide Gas Sensors by Finite Element Modeling and Infra-Red Imaging Characterization"

_materials, 2023, doi:10.3390/ma16186202_

Round 1
Reviewer 1 Report
1. In the novelty section in introduction should be briefer and the aim of the work should be empathized. Why metal oxides are superior than others in this case should be reported. 2. The fabrication is not clearly described here. Better elaboration and justification needed for such systems. 3. How ‘temperature distribution’ is significant here should be justified also. 4. Some articles have significance for your reference; (a) Das, P., Maruthapandi, M., Saravanan, A., Natan, M., Jacobi, G., Banin, E., & Gedanken, A. (2020). Carbon dots for heavy-metal sensing, pH-sensitive cargo delivery, and antibacterial applications. ACS Applied Nano Materials, 3(12), 11777-11790. (b) Das, P., Ganguly, S., Mondal, S., Bose, M., Das, A. K., Banerjee, S., & Das, N. C. (2018). Heteroatom doped photoluminescent carbon dots for sensitive detection of acetone in human fluids. Sensors and Actuators B: Chemical, 266, 583-593. (c) Das, M., & Roy, S. (2021). Polypyrrole and associated hybrid nanocomposites as chemiresistive gas sensors: A comprehensive review. Materials Science in Semiconductor Processing, 121, 105332.
N
Author Response
Author's Reply to the Review Report (Reviewer 1): Please see the attachment

Reviewer 2 Report
- Line 129, may you use × instead of x
- What is the unit of the color bar in Figure 3
- Line 197, do you want to write 1 105 like 1×105 or 1E05? then check the whole paper with the same format
- Commonly the gas sensor technology started to activate the surface by doping, or nano decorations than using a heating source to able to commercialize the room temperature gas sensors, for example, “Basyooni, M., Shaban, M. & El Sayed, A. Enhanced Gas Sensing Properties of Spin-coated Na-doped ZnO Nanostructured Films. Sci Rep 7, 41716 (2017). https://doi.org/10.1038/srep41716”. However, for your study, you intended to fabricate a new model for heaters with high heat distribution. May you investigate this issue and high your research question
- It seems that during the whole article, you are talking about gas sensors, however, there is not any data regarding the gas sensor, which gas you tested, sensitivity, response time, detection limit, or recovery time, …. Many other parameters needed to be collected for any gas sensor setup.
- If you need to focus on the heat that might be used in the gas sensor, you can mention these as potential applications by the end of the article. Because by reading the article, I want to see the real application of a gas sensor system, but nothing there. So it is better to remove any gas sensor, especially from the title unless you can provide a small test on a real gas sensor system theoretically or experimentally
- In the introduction part, the literature review about some gas sensor setups that used different heating systems has to be mentioned, and infra-red imaging systems.
- We need to compare your system performances with the others
Moderate editing of English language required
Author Response
Author's Reply to the Review Report (Reviewer 2): Please see the attachment.

Round 2
Reviewer 1 Report
I did not find the references mentioned by the reviewers. Those should be highlighted as per the suggestion.
Author Response
Authors’ response: Thank you for your suggestions.
Authors’ action:
We have updated the manuscript by adding additional information.
We have updated the manuscript by adding all additional references, as suggested.
I have attached a marked version to the manuscript to highlight any changes made.

Reviewer 2 Report
thank you for your report
It has to be revised
Author Response
Authors’ response: Thank you for your suggestions.
Authors’ action:
The manuscript was proofread by a near-native English speaker (someone who has lived in Australia for more than 10 years). We improved the language of the manuscript.
We have updated the manuscript by adding all additional references, as suggested.
I have attached a marked version to the manuscript to highlight any changes made.
